# Post-vaccination SARS-CoV-2 IgG spike antibody responses among clinical and non-clinical healthcare workers at a tertiary facility in Kenya

**Lucy W. Mwangi**[1], **Geoffrey Omuse**[2], **Rodney Adam**[2,3], **George Ong'ete**[4], **Cyrus Matheka**[2], **Patrick Mugaine**[2], **Shahin Sayed**[2], **Daniel Maina**[2]*

1 Research Division, Medical College East Africa, The Aga Khan University Hospital, Nairobi, Kenya,
2 Department of Pathology and Laboratory Medicine, The Aga Khan University Hospital, Nairobi, Kenya,
3 Department of Internal Medicine, The Aga Khan University Hospital, Nairobi, Kenya, 4 Occupational Safety and Health, The Aga Khan University Hospital, Nairobi, Kenya

* dmaina@aku.edu

## Abstract

### Introduction

Following the coronavirus disease 19 (COVID-19), caused by severe acute respiratory syndrome coronavirus-2 (SARS-CoV-2) infection, vaccination became the main strategy against disease severity and even death. Healthcare workers were considered high-risk for infection and, thus, were prioritised for vaccination.

### Methods

A follow-up to a SARS-CoV-2 seroprevalence study among clinical and non-clinical HCWs at the Aga Khan University Hospital, Nairobi, we assessed how vaccination influenced SARS-CoV-2 anti-spike IgG antibody responses and kinetics. Blood samples were drawn at two points spanning 6 to 18 months post-vaccination, and SARS-CoV-2 spike antibody levels were determined by enzyme-linked immunosorbent assay.

### Results

Almost all participants, 98% (961/981), received a second vaccine dose, and only 8.5% (83/981) received a third dose. SARS-CoV-2 spike IgG antibodies were detected in 100% (961/961) and 92.7% (707/762) of participants who received two vaccine doses, with the first and second post-vaccine test, respectively, and in 100% (83/83) and 91.4% (64/70) of those who received three vaccine doses at the first and second post-vaccine test, respectively. Seventy-six participants developed mild infections, not requiring hospitalisation even after receiving primary vaccination. Receiving three vaccine doses influenced the anti-spike S/Co at both the first (p<0.001) and second post-vaccination testing (p<0.001). Of those who tested SARS-CoV-2 positive, the anti-spike S/Co ratio was significantly higher than those who were seronegative at the first post-vaccine test (p = 0.001). Side effects were reported

**Funding:** The author(s) received no specific funding for this work.

**Competing interests:** The authors have declared that no competing interests exist.

by almost half of those who received the first dose, 47.3% (464/981), 28.9% (278/961) and 25.3% (21/83) of those who received the second and third vaccine doses, respectively.

## Discussion and conclusion

Following the second dose of primary vaccination, all participants had detectable anti-spike antibodies. The observed mild breakthrough infections may have been due to emerging SARS-CoV-2 variants. Findings suggest that although protective antibodies are induced, vaccination protected against COVID-19 disease severity and not necessarily infection.

## Introduction

Over 768.5 million people globally were confirmed to have Coronavirus disease-19 (COVID-19), and over 6.9 million resultant deaths occurred by July 2023. Kenya recorded 343,915 cases and 5,689 deaths by July 2023 (1). Four years since the beginning of the pandemic in December 2019, the full effects of the disease caused by severe acute respiratory syndrome coronavirus-2 (SARS-CoV-2) have yet to be determined. However, vaccination was a major prevention strategy against severe disease across the globe, with the World Health Organisation (WHO) rolling out the global initiative for rapid access to COVID-19 vaccines in early 2021. By July 2023, over 13 billion vaccine doses had been administered worldwide (172.7 doses per 100 population), 65.91 vaccine persons per 100 population receiving a complete primary series (2 doses), and 31.73 vaccine persons per 100 population receiving at least one booster or additional dose. Most of the world's unvaccinated population is from low- and middle-income countries (LMICs) [1]. A disparity in vaccination is noted in the WHO African region, which reported that only 32.6% of the population had completed the primary vaccination series. In Kenya, 23,750,431 vaccine doses had been administered by July 2023, with only 21.4% of the adult population having completed the primary vaccination series [2].

Vaccination influences antibody responses and improves immune protection. Indeed, heterologous boosting has resulted in more robust immune responses than homologous boosting [3–5]. Immunoglobulin G (IgG) antibodies are necessary for immune protection against SARS-CoV-2, and administering different vaccines may result in different antibody responses over time. Vaccination and infection-induced immunity resulted in somewhat a plateau in new COVID-19 infections; nonetheless, as the pandemic continued, new variants of SARS-CoV-2 emerged [6]. Healthcare workers were categorised as high-risk at the onset and during the pandemic and thus were prioritised for vaccination alongside other vulnerable populations such as older adults [7].

This study assessed how the demographics and different vaccines administered impacted SARS-CoV-2 anti-spike antibody signal-to-cut-off ratios (anti-S S/Co) and COVID-19 infections among clinical and non-clinical health care workers (HCWs) at the Aga Khan University Hospital, Nairobi (AKUHN).

## Materials and methods

This was a follow-up study of over 1,600 HCWs who had participated in an earlier pre-vaccine SARS-CoV-2 seroprevalence study. The study population and setting have been described previously (8). Clinical and non-clinical HCWs working within AKUHN who were vaccinated were included in this follow-up study. Recruitment of participants for this phase started on October 19, 2021, and ended on November 30, 2022.

## Sample collection and laboratory testing

Each participant had blood samples drawn at two-time points spanning 6–18 months after vaccination. Additional data on vaccination, possible side effects, and a history of new SARS-CoV-2 infections were captured in an already existing REDCap® (Vanderbilt University) database. SARS-CoV-2 spike antibodies were determined using the Wantai® total spike antibody kit (Beijing Wantai Biological Pharmacy Enterprise, China) on an automated ELISA and chemiluminescent platform respectively. The Wantai® kit has a manufacturer's reported sensitivity of 96.7% and a specificity of 97.5%. Results from these tests were expressed semi-quantitatively as a signal-to-cut-off ratio (S/Co) for both assays. A S/Co value $\geq 1.0$ was considered positive.

## Statistical analysis

We determined the kinetics of immune response (spike antibodies) following vaccination and factors that could impact the response, such as age, sex, prior SARS-CoV-2 infections (evidence-PCR, nucleocapsid antibodies) and the time interval between the first and subsequent vaccine doses, and the time between vaccinations and sample collection. We also considered the time interval between prior confirmed SARS-CoV-2 infections and sample collection. Antibody kinetics (at first and second post-vaccination samples) were analysed using the Wilcoxon signed-rank test. Associations between antibody response expressed as signal-to-cut-off (S/Co) ratios and demographic and clinical parameters were assessed using Chi-square statistics and regression analysis. The Kruskal-Wallis H test interrogated whether anti-S S/Co differed depending on vaccine type. Descriptive statistics are presented here as means and standard deviations (SD), medians and interquartile ranges (IQR), and proportions in tables and boxplots. Associations are presented as chi-square statistics and correlation coefficients, using a 5% alpha set to determine statistical significance ($p < 0.05$). STATA® 15 (StataCorp, Texas, USA) was used to perform the statistical analyses.

## Ethics statement

The study was approved by the Aga Khan University Scientific and Ethics Review Committee [Ref:2020/IERC-129(v6)]. All procedures were guided by research ethics principles: respect, beneficence, and justice. Written informed consent was obtained from all study participants.

## Results

### Participant characteristics

The initial seroprevalence study summarises the characteristics of the study participants [8]. Briefly, 1,631 clinical and non-clinical staff working at AKUH-N were included. Of these, 981 (60%) participated in this follow-up study, providing at least one post-vaccination blood sample, and 764 (47%) provided two samples. Table 1 summarises the characteristics of the 981 post-vaccination participants.

Almost all participants, 98%, received a second vaccination dose, while very few 83 (8.5%) received the third vaccine dose. Oxford/AstraZeneca® (ChAdOx1-S [recombinant] vaccine) was the vaccine commonly administered for both the first and second doses [788 (80.3%) and 790 (82.2%), respectively]. Fifty-two (62.7%) participants who received a third dose received the Pfizer-BioNTech® (BNT162b2) vaccine. Almost half (47.3%) of the participants who received the first vaccine dose indicated side effects other than having a sore arm at the vaccination site, 56% of whom were women. Of those who received the second and third (boost) vaccine doses, only 28.9% and 25.3% indicated side effects other than a sore arm, respectively.

**Table 1. Participant characteristics, type and frequency of vaccines administered at each dose.**

| | Dose 1 | Dose 2 | Dose 3 |
|---|---|---|---|
| **Vaccinated, n** | 981 | 961 | 83 |
| **Female, n [%]** | 554 [56.5] | 538 [56.0] | 43 [51.8] |
| **Male, n [%]** | 427 [43.5] | 423 [44.0] | 40 [48.2] |
| **Type of Staff, n [%]** | | | |
| AKUH-N | 826 [84.2] | 807 [84] | 67 [80.7] |
| Contracted | 155 [15.8] | 154 [16] | 16 [19.3] |
| **Age in years** | | | |
| 20–29 | 212 | 209 | 22 |
| 30–39 | 403 | 393 | 34 |
| 40–49 | 248 | 244 | 13 |
| 50–59 | 84 | 81 | 9 |
| 60+ | 21 | 21 | 4 |
| **Age not indicated** | 13 | 13 | 1 |
| **Type of Vaccine, n [%]** | | | |
| AstraZeneca | 788 [80.3] | 790 [82.2] | 7 [8.43] |
| Johnson & Johnson | 21 [2.14] | 1 [0.1] | - |
| Moderna | 122 [12.44] | 121 [12.59] | 24 [28.9] |
| Pfizer | 50 [5.09] | 49 [5.09] | 52 [62.65] |
| **Side Effects*** | | | |
| Yes n [%] | 464 [47.3] | 278 [28.9] | 21 [25.3] |

*Side effects other than a sore arm.
AstraZeneca = Oxford/AstraZeneca® (ChAdOx1-S [recombinant]; Johnson & Johnson = Jansen/Johnson & Johnson® Ad26.COV2.S; Moderna = Moderna® (mRNA-1273); Pfizer = Pfizer-BioNTech® (BNT162b2).

Of participants who reported side effects, the frequency of side effects was higher among participants who received ChAdOx1-S as the first (84.5%) and second (91.4%) doses. For the third dose, side effects were more frequent among participants who received BNT162b2 vaccine (61.9%). Some participants experienced two or more side effects. Additionally, women who received the second vaccine dose reported more than half of the side effects (55.7%). However, with the third booster dose, more men, 71.4%, reported having experienced an effect other than a sore arm. Table 2 summarises the side effects reported.

The median duration between the first and second vaccine doses was 72 days [interquartile range (IQR): 49–88], and between the second and third doses was 266 days [IQR: 222–308]. The median duration between the first vaccine dose and the initial post-vaccination antibody testing was 281 days [IQR: 255–304], while that between the second vaccine dose and the initial post-vaccination antibody testing was 208 days [IQR: 186–228]. Table 3 summarises the median durations between vaccinations and SARS-CoV-2 antibody testing.

## Breakthrough infections

Seventy-six out of 153 (50%) participants who developed symptoms compatible with COVID-19 and were tested after the parent SARS-CoV-2 seroprevalence study had a positive result, but none required hospitalisation. Seventy-five were tested by polymerase chain reaction (PCR), while one was tested using a rapid antigen test. All 76 had received at least two vaccine doses, while of the 75 PCR tested, five received three vaccine doses (2 Moderna® mRNA vaccine; 3 Pfizer-BioNTech®). The median duration between the second vaccine dose and post-

**Table 2. Frequency of side effects of vaccination.**

| | Dose 1 (n = 981) | | | | Dose 2 (n = 961) | | | Dose 3 (n = 83) | | |
|---|---|---|---|---|---|---|---|---|---|---|
| | Astra-Zeneca (n = 788) | J&J (n = 21) | Moderna (n = 122) | Pfizer (n = 50) | Astra-Zeneca (n = 790) | Moderna (n = 121) | Pfizer (n = 49) | Astra-Zeneca (n = 7) | Moderna (n = 24) | Pfizer (n = 52) |
| Side Effect | | | | | | | | | | |
| Headache | 348 | 6 | 36 | 10 | 194 | 17 | 6 | - | 6 | 12 |
| Fever | 291 | 11 | 40 | 9 | 148 | 21 | 8 | 1 | 6 | 11 |
| Malaise | 140 | 6 | 27 | 7 | 71 | 9 | 3 | 1 | 3 | 6 |
| Muscle Pain | 253 | 5 | 26 | 9 | 134 | 8 | 5 | - | 3 | 3 |
| Other* | 21 | 1 | 6 | - | 4 | - | - | - | - | 2 |

*Other side effects reported after the first vaccine dose included gastrointestinal effects such as abdominal pain and discomfort, nausea and vomiting, chest pain, dizziness, facial rash, chills and clotting during menses. | Other side effects reported after the second vaccine dose included nausea and vomiting, chills, mild chest pain, and clotting during menses. | Other side effects reported after the third vaccine dose included abdominal pain, vomiting and mild chest pain.

Astra-Zeneca = Oxford/AstraZeneca® (ChAdOx1-S [recombinant]; J&J = Jansen/Johnson & Johnson® Ad26.COV2.S; Moderna = Moderna® (mRNA-1273); Pfizer = Pfizer-BioNTech® (BNT162b2).

vaccination SARS-CoV-2 polymerase chain reaction (PCR) testing was 149 [IQR: 85–179] days. The median duration between pre- and post-vaccination SARS-CoV-2 PCR testing was 360 days [IQR: 293–384] (Table 3).

## Anti-spike antibody response

Two post-vaccination blood samples were collected separately for anti-SARS-CoV-2 IgG testing. These two tests had a median duration of 238.5 [IQR: 202–267.5] days (approximately eight months) between them. Of participants who received two vaccine doses, or the Jansen/Johnson & Johnson® Ad26.COV2.S vaccine, which was administered as a single dose, SARS-CoV-2 spike IgG antibodies (anti-S antibodies) were detected among all 100% (961/961) at the first post-vaccination testing and in 92.8% (707/762) at the second post-vaccination testing. Of participants who received three vaccine doses, anti-S antibodies were detected in 100% (83/83) at the first post-vaccination testing and on 91.4% (64/70) after the second post-vaccination

**Table 3. Median time interval between vaccination doses and post-vaccine sample collection.**

| | Median [IQR] (Days) | Median (Months) |
|---|---|---|
| Doses 1 and 2 | 72 [49 to 88] | 2.3 |
| Doses 1 and 3 | 317.5 [292 to 391] | 10.2 |
| Doses 2 and 3 | 265.5 [222 to 308] | 8.5 |
| Dose 1 and post-vaccine sample 1 | 281 [255 to 304] | 9.0 |
| Dose 1 and post-vaccine sample 2 | 519 [481 to 554.5] | 16.7 |
| Dose 2 and post-vaccine sample 1 | 208 [186 to 228] | 6.7 |
| Dose 2 and post-vaccine sample 2 | 441 [411 to 471] | 14.2 |
| *Dose 3 and post-vaccine sample 1 | 39.5 [-16 to 88] | 1.3 |
| *Dose 3 and post-vaccine sample 2 | 173.5 [144 to 230] | 5.6 |
| Post-vaccine samples 1 and 2 | 238.5 [202 to 267.5] | 7.6 |
| Post-vaccine sample 2 and post-vaccine test | 149 [85 to 179] | 4.8 |
| Pre-vaccine and post-vaccine test | 360 [293 to 384] | 11.6 |

*Sample collection was prior to vaccine dose 3 for some participants.

**Table 4. SARS-CoV-2 anti-spike antibody signal-to-cut-off (S/Co) ratio following primary and booster vaccination.**

| | | VACCINE DOSE 2 | | | | VACCINE DOSE 3 | | |
|---|---|---|---|---|---|---|---|---|
| | | AstraZeneca | J & J* | Moderna | Pfizer | AstraZeneca | Moderna | Pfizer |
| POST VACCINE SAMPLE 1 | Positive (n) | 790 | 1 | 121 | 49 | 7 | 24 | 52 |
| | S/Co Ratio | | | | | | | |
| | Mean | 119.03 | 127.53 | 113.07 | 120.58 | 129.36 | 126.31 | 130.70 |
| | sd | 27.88 | | 23.12 | 23.33 | 19.82 | 21.51 | 22.83 |
| POST VACCINE SAMPLE 2 | Positive (n) | 598 | - | 87 | 22 | 6 | 20 | 38 |
| | S/Co Ratio | | - | | | | | |
| | Mean | 23.94 | | 38.46 | 23.62 | 39.22 | 79.43 | 82.42 |
| | sd | 34.40 | | 37.67 | 33.83 | 43.79 | 27.21 | 30.86 |
| | Negative (n) | 52 | 1 | 2 | - | 1 | 1 | 4 |
| | S/Co Ratio | | | | - | | | |
| | Mean | 0.65 | 0.38 | 0.57 | | 0.91 | 0.85 | 0.43 |
| | sd | 0.26 | | 0.12 | | | | 0.32 |

Mean and sd–to 2 decimal point; sd = standard deviation | All post-vaccine sample 1 were positive for anti-spike IgG. | *One participant received J&J given as a single vaccine dose.

AstraZeneca = Oxford/AstraZeneca® (ChAdOx1-S [recombinant]; J&J = Jansen/Johnson & Johnson® Ad26.COV2.S; Moderna® = Moderna (mRNA-1273);
Pfizer = Pfizer-BioNTech® (BNT162b2).

testing respectively. Table 4 summarises the frequency of anti-S positivity and mean S/Co ratio of anti-S antibodies among participants after receiving second and booster vaccine doses using different vaccine types.

Among the 153 participants who were re-tested for SARS-CoV-2 by PCR, those who had a positive PCR had a higher anti-S S/Co ratio at the first post-vaccine test than those with negative PCR tests (Mann-Whitney U test, p = 0.001), irrespective of sex (Fig 1A and 1C). However, no difference between positive versus negative PCR results was found at the second post-vaccination testing (p = 0.791) (Fig 1B). Simple regression was performed to test whether the anti-nucleocapsid (anti-N) result could significantly predict the anti-S S/Co ratio. Indeed, anti-N predicted anti-S S/Co ratio at both the first and second post-vaccine test (β = 7.042, p<0.001 and β = 47.122, p<0.001 respectively). By age category, the median anti-S S/Co difference was observed only in the 20–29 age category (p<0.001) at the first post-vaccination testing (Fig 1D). At the second post-vaccine testing, there was no difference in median anti-S S/Co ratio by sex between those who tested positive and negative for SARS-CoV-2 [male p = 0.852, female p = 0.828]. No difference was observed by age group between those who tested positive or negative for SARS-CoV-2 at re-test (S1 Fig).

The Mann-Whitney U-test was performed to assess whether the number of vaccine doses influenced the anti-S S/Co ratio. At the first post-vaccination testing, among those who received a second vaccine dose, the median S/Co ratio was not significantly different between those who received a second vaccine dose and those who did not (p = 0.32) (Fig 2A). However, a difference trending to significance was observed at the second post-vaccination testing (p = 0.052) (Fig 2B). The anti-S S/Co did not differ with sex or age category at the first post-vaccine testing between those who did not receive a second vaccine dose and those who did (S2 Fig). Neither were there any significant differences at the second post-vaccination testing (S2 Fig). Receiving three vaccine doses influenced the anti-S S/Co at both the first (p<0.001) and second post-vaccination testing (p<0.001) (Fig 2C and 2D). Among female participants, the median anti-S S/Co ratio at the first post-vaccine test was significantly higher among those

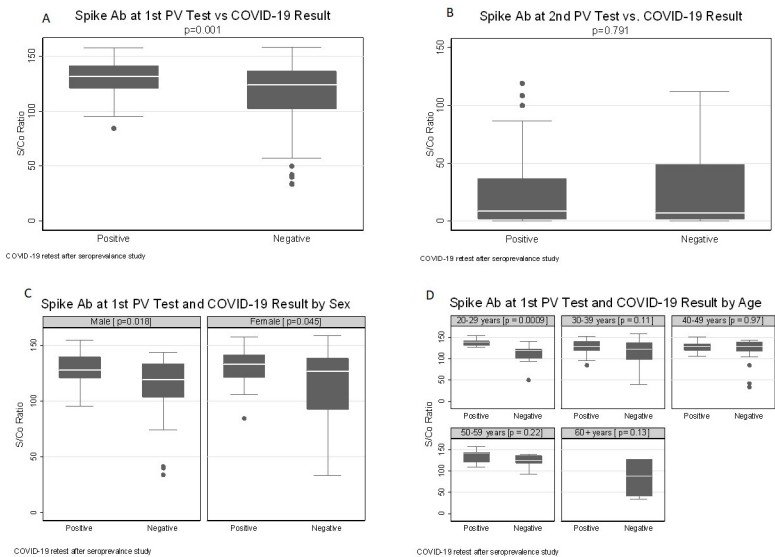

**Fig 1. Anti-spike antibody response based on COVID-19 result, sex and age.** Box plots of anti-Spike IgG S/Co ratio at first (A) and second (B) post-vaccination test between those with positive and negative SARS-CoV-2 test results. Anti-Spike IgG S/Co ratio at first and second post-vaccination testing by SARS-CoV-2 result by sex (C) and age category [n = 968] (D). SARS-CoV-2 PCR re-test was performed after the seroprevalence study and initiation of vaccination | PV = post vaccination.

who received a third dose (p<0.001). With the 30–39-year age category, the median anti-S-S/Co ratio was higher among those who received a third vaccine dose than those who did not (p = 0.005) (Fig 3A and 3B). At the second post-vaccination test, the median anti-S S/Co ratio was higher among those who received a third dose among both male (p<0.001) and female (p<0.001) participants (Fig 3C). By age, those who received a third vaccine dose showed a difference in median anti-S S/Co at the second post-vaccination testing in all but the 50–59 age group [p<0.001 in the 20–29 and 30–39 age groups, p = 0.022 in the 49–59 age group, and p = 0.030 in the over-60-year age group (Fig 3D).

Linear regression established that the duration between the first vaccine dose and the first post-vaccination test predicted the anti-S S/Co levels [vaccine dose 1: F (1,978); p = 0.03]. In addition, the duration from the second vaccine dose to the first post-vaccination test also predicted the anti-S S/Co levels [F (1,958); p = 0.004]. In contrast, for the second post-vaccination testing, only the time duration from the third vaccination dose was predictive of up to 66% of the variability of the anti-S S/Co ratio [F (1,68); p<0.001]. Among those who received only one vaccine dose, the anti-S S/Co differed by vaccine type [First post-vaccination test: Chi-square = 19.773 (3d.f), p<0.001. Second post-vaccination test: Chi-square 31.400 (3 d.f), p<0.001]. The median anti-S S/Co ratio also differed by vaccine type given at the second vaccination dose with either testing [First post-vaccination testing: Chi-square = 17.558 (3 d.f), p<0.001). Second post-vaccination testing: Chi-square = 27.5 (3 d.f), p<0.001]. However, the median anti-S S/Co ratio did not differ by vaccine type with the third vaccination dose at either testing [First post-vaccination test: Chi-square = 1.047 (2 d.f), p = 0.592. Second post-vaccination test: Chi-square = 5.321 (2d.f), p = 0.07] (S3 Fig).

Linear regression established that the duration between the first post-vaccination test and the first and second vaccination doses predicted the anti-S S/Co ratio [vaccine dose 1: F (1,978); p = 0.03]. In addition, the duration from the second vaccine dose to the first post-

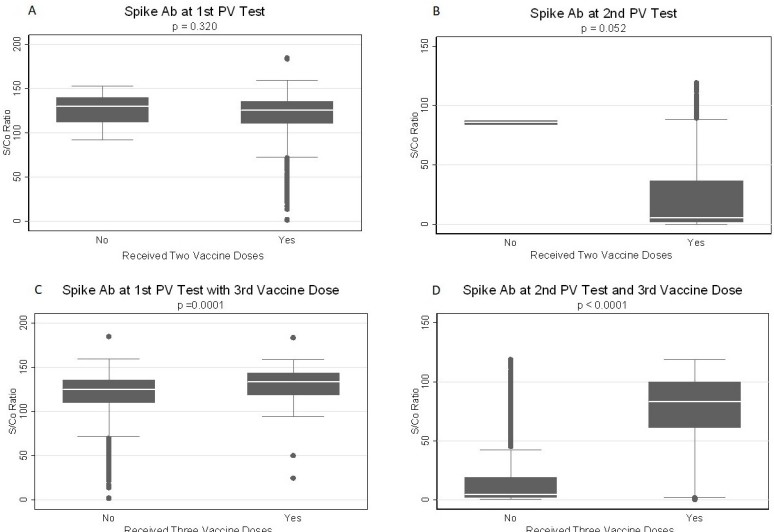

**Fig 2. Anti-spike antibody response at second and third vaccine doses.** Box plots of anti-spike IgG S/Co ratio at first (A) and second (B) post-vaccination test between those who received a second vaccine dose and those who did not receive two doses. Anti-Spike IgG S/Co ratio at first (C) and second (D) post-vaccination testing between participants receiving three vaccine doses and those not receiving three doses. | PV = post vaccination.

vaccination test also predicted the anti-S S/Co ratio [$F_{(1,958)}$; p = 0.004]. In contrast, for the second post-vaccination testing, only the time duration from the third vaccination dose was predictive of up to 66% of the variability of the anti-S S/Co ratio [$F_{(1,68)}$; p<0.001]. Among those who received only one vaccine dose, the anti-S S/Co differed by vaccine type [First post-vaccination test: Chi-square = 19.773 (3d.f), p<0.001. Second post-vaccination test: Chi-square

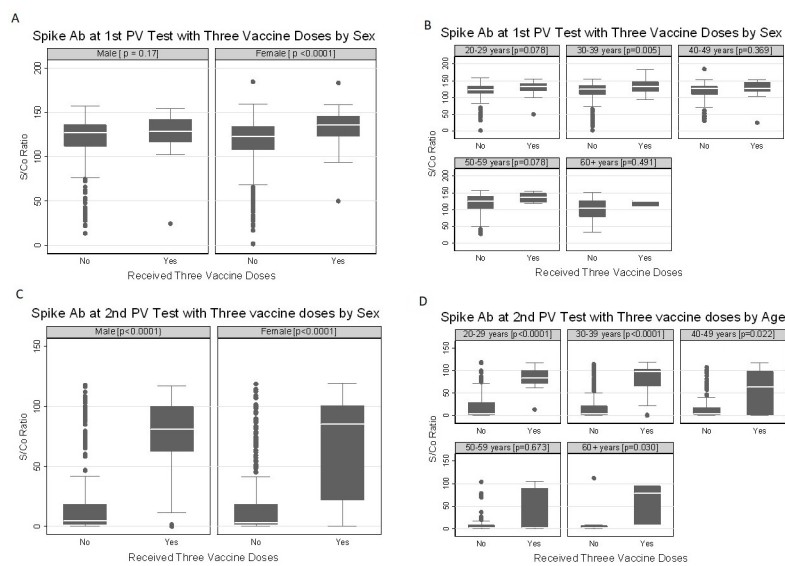

**Fig 3. Anti-spike antibody response with third vaccine dose.** Box plots of anti-spike IgG S/Co ratio at first and second post-vaccination test among those who received a third vaccine dose and those who did not, by sex (A and C) and age (B and D) | PV = post vaccination [n = 968].

31.400 (3 d.f), p<0.001]. The median anti-S S/Co ratio also differed by vaccine type at the second vaccination dose with either testing [first post-vaccination testing: Chi-square = 17.558(3 d.f), p<0.001; second post-vaccination testing: Chi-square = 27.5 (3 d.f), p<0.001). However, the median anti-S S/Co ratio did not differ by vaccine type with the third vaccination dose at either testing [first post-vaccination test: Chi-square = 1.047 (2 d.f), p = 0.592; second post-vaccination test: Chi-square = 5.321(2d.f), p = 0.07] (S3 Fig).

## Discussion

Vaccination was the main strategy against COVID-19 disease severity [9]. Kenya initiated vaccination in early 2021, with the ChAdOx1-S recombinant vaccine by Oxford/AstraZeneca® being the first to be rolled out following the global COVAX initiative. Frontline workers, including HCWs, security personnel, and older adults, were prioritised to receive vaccination [7]. As such, almost all participants in our study who received the primary vaccination (defined as two doses of either the Oxford/AstraZeneca® ChAdOx1-S, Pfizer-BioNTech® BNT162b2, or Moderna® mRNA vaccine or one dose of the Johnson & Johnson® Ad26. CoV2.S vaccine) received the ChAdOx1-S recombinant vaccine. Primary vaccine coverage among our study cohort was high (over 97%). Similar high vaccination coverage has been shown in a study from Kenya where 92% of HCWs had been partially or fully vaccinated [10].

In contrast to the Kenyan study, an analysis of COVID-19 vaccination in the WHO African region found that only 40.9% of HCWs had completed the primary vaccination series by the end of 2022 [11]. Women comprise approximately 60% of the HCWs at AKUH-N; therefore, it was not surprising that more women (56.5%) than men participated in our study. This is similar to observations elsewhere that HCWs, observed to mainly be women, were prioritised for vaccination and that women made up the majority of participants in vaccine non-randomised clinical trials [12]. Side effects frequently reported among participants who received the ChAdOx1-S recombinant vaccine were likely because most participants received this type following the rollout of vaccination in Kenya. Adverse events more frequently reported among female participants supported previous findings indicating that women were more likely to report adverse and local side effects than men [13,14].

Generally, a history of SARS-CoV-2 infection has been associated with a lower risk of new infection for more than six months [15,16]. Vaccinated individuals, similarly, have a reduced risk of developing severe disease [16]. Indeed, immunoglobulin (IgG) against the spike antibody remains stable for over six months post a natural infection [17]. Our study observed 76 breakthrough infections among this cohort of HCWs who had received the primary vaccination series; six of them also had documented SARS-CoV-2 prior to vaccination, but none required hospitalisation. The median time to a positive result was less than six months after the completion of primary vaccination, suggesting the waning of anti-SARS-CoV-2 antibodies. These breakthrough infections could result from infection by different variants of SARS-CoV-2. Notably, the Omicron variant was the predominant circulating strain [6].

Previous studies have shown similar findings among fully vaccinated HCWs. Bergwerk and colleagues, for instance, showed that in a cohort of 1,497 HCWs, 2.6% experienced breakthrough infections, which were mostly asymptomatic. However, they also observed that some symptoms lasted beyond six weeks [18]. Locally, there is a paucity of data on SARS-CoV-2 reinfection post-vaccination; studies from elsewhere have suggested that reinfections may be more prevalent among HCWs than the general population [19,20]. Our findings support the conclusion that anti-spike antibodies are induced by vaccination and are protective against severe COVID-19 disease in the short term. However, the waning antibody levels indicate that

longer-term follow-up will be required to determine the duration of protection from severe COVID-19 disease [21].

Our study showed that anti-spike antibody titres were higher among the vaccinated with a previous SARS-CoV-2 infection, irrespective of sex. This finding is similar to observations that natural infection prior to vaccination boosted anti-spike antibodies among HCWs in Romania [22] and South Africa [23] as well as in a non-HCW population [24]. In addition, our study found that age influenced anti-spike antibody titres for those between 20 and 29 years old. However, we did not make similar observations at the second post-vaccination test, which averaged more than seven months after the first post-vaccination test and 14 months after completion of the primary vaccination among participants. Indeed, it has previously been shown that time since vaccination is associated with SARS-CoV-2 antibody attrition [25].

We acknowledge the limitations of this study. First, although we had a relatively large sample size, some participants received full primary vaccination before the first sample collection; therefore, some of the measured antibody levels also accounted for full vaccination or even boosting. Nonetheless, we were able to show that the number of doses and duration since vaccination influenced the antibody titres of our population. Secondly, 99% of the participants who received the full primary series received homologous immunisation, making it impossible to assess the effect of heterologous prime-boost vaccination. Future research should incorporate heterologous vaccination participants to further evaluate the longevity and efficacy of antibody responses, especially with the emergence of new SARS-CoV-2 strains.

## Conclusion

Overall, this study shows that anti-spike IgG is detectable for about one year after the primary vaccination series is completed, although the levels do wane with time. Breakthrough infections underscore that vaccination does not necessarily prevent SARS-CoV-2; however, infection may not require hospitalisation.

## Supporting information

**S1 Fig. Anti-spike IgG S/Co ratio at second post-vaccination testing and COVID-19 test.** Stratified by sex (S1a) and age category (S1b) [age n = 968].
(JPG)

**S2 Fig. Anti-spike IgG S/Co ratio at first and second post-vaccine testing with second vaccine does.** Stratified by sex (S2a and S2c) and age category (S2b and S2d) [age n = 968].
(JPG)

**S3 Fig. Anti-spike IgG S/Co ratio based on vaccine type and number of doses.** Dose 1 (S3a, S3b), Dose 2 (S3c, S3d) and Dose 3 (S3e, S3f).
(JPG)

## Acknowledgments

Appreciation to David Kawalya and Assumpta Chege for logistical support, Augustine Gitonga for assisting with copy-editing and to the AKUH-N for facilitating the study.

## Author Contributions

**Conceptualization:** Daniel Maina.

**Data curation:** Lucy W. Mwangi, Daniel Maina.

**Formal analysis:** Lucy W. Mwangi.

**Investigation:** George Ong'ete, Cyrus Matheka, Daniel Maina.

**Methodology:** Geoffrey Omuse, Rodney Adam, Shahin Sayed, Daniel Maina.

**Project administration:** Patrick Mugaine.

**Software:** Lucy W. Mwangi.

**Supervision:** Rodney Adam, George Ong'ete.

**Validation:** Lucy W. Mwangi, Daniel Maina.

**Writing – original draft:** Lucy W. Mwangi.

**Writing – review & editing:** Lucy W. Mwangi, Geoffrey Omuse, Rodney Adam, George Ong'ete, Cyrus Matheka, Shahin Sayed, Daniel Maina.

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
