## [Decision Letter · Decision Letter 0]

3 Jan 2024

PONE-D-23-40445Post-Vaccination SARS-CoV-2 IgG Spike Antibody Responses among Clinical and Non-Clinical Healthcare Workers at a Tertiary Facility in Kenya.PLOS ONE

Dear Dr. Maina,

Thank you for submitting your manuscript to PLOS ONE. After careful consideration, we feel that it has merit but requires some minor revisions to fully meet PLOS ONE’s publication criteria. Therefore, we invite you to submit a revised version of the manuscript that addresses the points raised during the review process.

ACADEMIC EDITOR: There are some suggestions made by the Reviewers that you may need to attend to consider. Kindly attend to these and respond appropriately where necessary.

We look forward to receiving your revised manuscript.

Kind regards,

Olatunji Matthew Kolawole, Ph.D.

Academic Editor

PLOS ONE

Journal Requirements:

3. We note that your Data Availability Statement is currently as follows: "All relevant data are within the manuscript and its Supporting Information files."

Reviewers' comments:

Reviewer's Responses to Questions

**Comments to the Author**

1. Is the manuscript technically sound, and do the data support the conclusions?

Reviewer #1: Yes

Reviewer #2: Yes

Reviewer #3: Yes

Reviewer #4: Yes

2. Has the statistical analysis been performed appropriately and rigorously? 

Reviewer #1: Yes

Reviewer #2: Yes

Reviewer #3: Yes

Reviewer #4: Yes

3. Have the authors made all data underlying the findings in their manuscript fully available?

Reviewer #1: Yes

Reviewer #2: Yes

Reviewer #3: Yes

Reviewer #4: Yes

4. Is the manuscript presented in an intelligible fashion and written in standard English?

Reviewer #1: Yes

Reviewer #2: Yes

Reviewer #3: Yes

Reviewer #4: Yes

5. Review Comments to the Author

Reviewer #1: Nice and educative work. It will be nice to also include the benefits of Random Binding Domain/N terminal domain (RBD-NTD vaccines as promising booster vaccines against different variants of SARS COV 2.

REF:

Montgomerie I, Bird TW, Palmer OR, Mason NC, Pankhurst TE, Lawley B, Hernández LC, Harfoot R, Authier-Hall A, Anderson DE, Hilligan KL, Buick KH, Mbenza NM, Mittelstädt G, Maxwell S, Sinha S, Kuang J, Subbarao K, Parker EJ, Sher A, Hermans IF, Ussher JE, Quiñones-Mateu ME, Comoletti D, Connor LM; VAANZ Group. Incorporation of SARS-CoV-2 spike NTD to RBD protein vaccine improves immunity against viral variants. iScience. 2023 Apr 21;26(4):106256. doi: 10.1016/j.isci.2023.106256. Epub 2023 Feb 20. PMID: 36845030; PMCID: PMC9940465.

Reviewer #2: This study by Daniel Maina, et al., assessed how vaccination influenced SARS-CoV-2 anti-spike IgG antibody responses and kinetics among over 1,600 clinical and non-clinical HCWs who had participated in an earlier pre-vaccine 78 SARS-CoV-2 seroprevalence study at the Aga Khan University Hospital, Nairobi, Kenya. Recruitment of participants for the study started on 19th October 2021 and ended on 30th 79 80 November 2022.

Blood samples were drawn at two time points spanning 6 to 18 months post vaccination and SARS-CoV-2 spike antibody levels determined using the Wantai® total spike antibody kit on an enzyme-linked-immunosorbent-assay (ELISA) platform.

Antibody kinetics were analyzed using the Wilcoxon signed rank test. Associations between antibody response expresses as a signal-to-cut-off (S/Co) ratio and demographic and clinical parameters were assessed using Chi-square statistics and regression analysis. Kruskal-Wallis H test interrogated whether anti-S S/Co differed depending on vaccine type. Descriptive statistics are presented as means and standard deviation (SD), medians and interquartile range (IQR), and proportions in tables and boxplots. Associations are presented as chi-square statistics and correlation of coefficients; a 5% alpha was set to determine statistical significance (p<0.05) STATA® 15 104 (Stata Corp, Texas, USA) was used to perform the statistical analyses.

Results revealed that SARS-CoV-2 spike IgG antibodies were detected in 100% (961/961) and 92.7% (707/762) and of participants who received two vaccine doses with the first and second post-vaccine test respectively; and in 100% (83/83) and 91.4% (64/70) of those who received three vaccine doses at the first and second post-vaccine test respectively. Seventy-six HCWs became mildly infected even after receiving primary vaccination. Receiving three vaccine doses influenced the anti-spike S/Co at both the first (p<0.001) and second post-vaccination testing (p<0.001). Of those who tested SARS-CoV-2 positive, anti-spike S/Co ratio was significantly higher than among those who had tested negative at the first post-vaccine test (p=0.001). Side effects were reported by almost half of those who received the first dose 47.3% (464/981), 28.9% (278/961) and 25.3% (21/83) of those who received the second and third vaccine doses respectively.

Dependent (outcome) variables: SARS-CoV-2 spike IgG antibodies

Independent (predictors) variables: age, sex, prior SARS-CoV-2 infections, time interval between the first and subsequent vaccine doses, time between vaccinations and sample collection, and time interval between prior confirmed SARS-CoV-2 infections and sample collection.

The authors made all data underlying the findings fully available.

The data was also analyzed using both descriptive and inferential statistics which were rigorous and appropriate.

Discussions of the results were robust, citing similar studies conducted both within and outside Kenya and the continent.

Conclusions are in line with the findings

Writing quality and clarity: Satisfactory

Other observations:

1. Limitations of the study: The authors did well to mention the limitations of the study. However, in view of these limitations, what are their recommendations for future research in this area?

2. Inclusion/exclusion criteria not clearly explained.

Reviewer #3: The paper quality might be improved if the following comments are considered.

1. In table 1, gender is only showing values for the females. It is better to include the date for the male gender as well and do not leave your audience to assume the population are only binary and hence the balance is that of males.

2. The limitation of the study stated under the discussion section should be brought out as a separate section probably after the conclusion of the article.

Reviewer #4: This Research Project aligned with the principles of research and publication ethics. Data for the research are readily available. Though I would have appreciate that findings can find a correlation between the high igG spike in the post vaccination and long term immunity against COVID-19.

6. PLOS authors have the option to publish the peer review history of their article (what does this mean?). If published, this will include your full peer review and any attached files.

Reviewer #1: **Yes: **Ado Garba Abubakar

Reviewer #2: **Yes: **Haruna Ismaila ADAMU, MBBS; MPH; PhD

Reviewer #3: No

Reviewer #4: **Yes: **Emmanuel Adamolekun

---

## [Author Response · Author response to Decision Letter 0]

26 Jan 2024

We resubmit our manuscript titled: Post-Vaccination SARS-CoV-2 IgG Spike Antibody Responses among Clinical and Non-Clinical Healthcare Workers at a Tertiary Facility in Kenya. We thank the reviewers a for their comments after reviewing our manuscript. We submit the revised manuscript for publication; together with the responses to the editor and reviewers, highlighted in blue.

Editor’s comments:

1. Ensure that the manuscript meets PLOS ONE’s style requirements.

Yes, the manuscript has been edited according to the guidelines.

2. Thoroughly copyedit the manuscript for language, spelling, and grammar.

With assistance from a librarian, Mr. Augustine Gitonga, the authors copy-edited the manuscript for grammar, spelling, and flow.

3. Data Availability Statement.

Relevant data are within the manuscript and its Supporting Information file. 

4. Review reference list.

Reference list reviewed.

Reviewers Comments:

Reviewer #1: Nice and educative work. It will be nice to also include the benefits of Random Binding Domain/N terminal domain (RBD-NTD vaccines as promising booster vaccines against different variants of SARS COV 2.

Thank you.

True. RBD-based vaccines present certain benefits against some emerging strains of SARS-CoV-2, therefore promising for vaccine boosting. Our study however was not designed to evaluate the efficacy of boosting mechanisms for the vaccines used for the booster dose.

Reviewer #2:

Thank you for your comments.

a. The authors did well to mention the limitations of the study. However, in view of these limitations, what are their recommendations for future research in this area?

This has been addressed – Clean copy, page 11

b. Inclusion/Exclusion criteria not clearly explained.

This has been edited to clarify – Clean copy, page 3

Reviewer #3

a. In table 1, gender shows values of females. It is better to include the data for the male gender as well and do not leave audience to assume the population are only binary and hence the balance is that of males.

This has been addressed – Clean copy, page 5

b. The limitation of the study stated under discussion section should be brought out in a separate section probably after the conclusion of the article.

Thank you. The authors however feel that this can be retained as part of the discussion.

Reviewer #4

This Research Project aligned with the principles of research and publication ethics. Data for the research are readily available. Though I would have appreciate that findings can find a correlation between the high igG spike in the post vaccination and long term immunity against COVID-19.

All participants had received at least one vaccine dose before enrolment. The study was not designed to establish anti-spike levels prior to vaccination (with natural infection only), and we are not able to distinguish to correlate antibody levels due to vaccination to long term-immunity against SARS-CoV-2.

---

## [Editor Report · Decision Letter 1]

8 Feb 2024

Post-Vaccination SARS-CoV-2 IgG Spike Antibody Responses among Clinical and Non-Clinical Healthcare Workers at a Tertiary Facility in Kenya.

PONE-D-23-40445R1

Dear Dr. Maina,

We’re pleased to inform you that your manuscript has been judged scientifically suitable for publication and will be formally accepted for publication once it meets all outstanding technical requirements.

Kind regards,

Olatunji Matthew Kolawole, Ph.D.

Academic Editor

PLOS ONE
---

## [Editor Report · Acceptance letter]

25 Mar 2024

PONE-D-23-40445R1 

PLOS ONE

Dear Dr. Maina, 

I'm pleased to inform you that your manuscript has been deemed suitable for publication in PLOS ONE. Congratulations! Your manuscript is now being handed over to our production team.

Kind regards, 

on behalf of

Dr. Olatunji Matthew Kolawole 

Academic Editor

PLOS ONE